# Attitudinal Acceptance of Intimate Partner Violence and Mental Health Outcomes for Female Survivors in Sub-Saharan Africa

**DOI:** 10.3390/ijerph18105099

**Published:** 2021-05-12

**Authors:** Reine-Marcelle Ibala, Ilana Seff, Lindsay Stark

**Affiliations:** 1Weill Cornell Medical College, 1300 York Avenue, New York, NY 10065, USA; rwi4001@med.cornell.edu; 2George Warren Brown School, Washington University in St. Louis, Campus Box 1196, 1 Brookings Drive, St. Louis, MO 63130, USA; seff@wustl.edu

**Keywords:** mental health, sub-Saharan Africa, intimate partner violence, attitudes, suicide ideation, adolescents

## Abstract

While current literature evidences a strong association between gender-based violence exposure and adverse mental health outcomes, few studies have explored how attitudinal acceptance of intimate partner violence (IPV) might impact this relationship. This analysis employed data from 13–24-year-old females as part of the Violence Against Children Surveys in Nigeria, Uganda, and Malawi. Mental health status, defined by the Kessler Screening Scale for Psychological Distress, and suicide ideation served as outcome measures. Predictors of interest included lifetime experiences of IPV and attitudinal acceptance of IPV. Country-stratified logistic and ordinary least squares regressions were used to predict outcomes and included interactions between violence exposure and attitudinal acceptance of IPV. Violence exposure was associated with increased symptoms of mental distress and increased suicide ideation in all countries. Among those who experienced IPV, exhibiting attitudinal acceptance of IPV was associated with improved mental health in Nigeria and Malawi. IPV tolerance conferred lower odds of suicide ideation following IPV exposure in Nigeria. The findings suggest that programs aiming to reduce attitudinal acceptance of IPV must consider how these changes may interact with women’s exposure to IPV.

## 1. Introduction

Gender-based violence (GBV) is a globally pervasive phenomenon that poses exacerbated risks of mental distress, substance use disorder, sexually transmitted infection, and death for women and girls [1]. According to the World Health Organization (WHO), one in three women globally has been a victim of physical and/or sexual violence by intimate partners and/or sexual violence by non-partners [2]. Although perpetration of GBV transcends socioeconomic, educational, religious, and cultural lines, its prevalence is especially pronounced in sub-Saharan Africa, where accounts of intimate partner violence (IPV) amongst ever-married/partnered women aged 15–49 years old, for instance, supersede worldwide averages by 22% [3,4]. Moreover, while low- and middle-income countries (LMICs) carry a greater burden of GBV, research on human rights violations in these settings, especially for adolescent girls, disproportionately lags behind efforts in high-income countries [3,5].

Adolescent girls are particularly vulnerable to GBV, their age and social status amplifying power imbalances and further limiting their autonomy [1,2,3,5,6]. The physical and mental toll of GBV greatly impacts survivors’ behaviors and lifetime health risks. Evidence suggests that experiences of GBV, including IPV, are correlated with elevated risks of liver disease, kidney dysfunction, arthritis, cardiovascular events, and HIV/AIDS [7,8]. Furthermore, in addition to exhibiting increased rates of substance use disorder, women exposed to GBV express greater incidence of anxiety disorders such as panic disorders and generalized anxiety disorders, social phobias and post-traumatic stress disorder (PTSD), and mood disorders including bipolar disorder, major depressive episodes, and dysthymia [9]. The correlation persists with self-harm practices, as the literature evidences a positive relationship between reports of emotional violence and suicide ideation, as well as general GBV occurrence and suicide attempts [10].

Researchers posit that a state of “mental defeat,” caused by abject loss of autonomy and personal power, and sentiments of self-isolation subsequent to trauma may be responsible for increased mental health symptoms [11]. According to a study examining suicide ideation and attempts in survivors of trauma, elevated levels of hopelessness and defeat correlate positively with suicidal behavior, regardless of PTSD symptomology [12]. Respondents diagnosed with PTSD demonstrate a similar positive correlation with increased suicidal behavior when expressing greater degrees of entrapment. Panagioti et al. [12] ascribe these effects to appraisal constructs defined by the Cry of Pain and Schematic Appraisal (SAMS) models of suicide. In keeping with these paradigms, survivors expressing heightened levels of hopelessness are more likely to harbor negative expectations of the future which contribute to the development of a defeated/entrapped mentality. The ensuing decrease in self-efficacy, along with the inability to perceive a feasible way out, draw survivors to self-harm and suicidal behavior as a means of escape [12].

The literature has also identified a bidirectional relationship between violence victimization and IPV tolerance. IPV tolerance, or attitudinal acceptance of violence, is defined as the belief that physical or emotional violence subjected against a romantic partner can be justified [13]. In a 2018 analysis of male and female attitudes towards IPV from 49 nationally representative demographic and health surveys, 36% of LMIC respondents demonstrated IPV-tolerant beliefs. While IPV-tolerant attitudes were least pervasive in Latin America and the Caribbean (11.87%), regional averages of IPV acceptance in sub-Saharan Africa (38.40%) came second only to those in South/South-East Asia (46.78%). Amongst sub-Saharan African countries, respondents in Malawi expressed the least IPV-tolerant beliefs (12.57%); respondents in Guinea (92.06%) expressed the highest levels of IPV tolerance [14]. Attitudinal acceptance of violence has been found to be associated with a higher likelihood of perpetrating violence among men and an elevated risk of victimization among women [15,16]. Conversely, survivors of GBV subjected to increasingly severe acts of violence and abuse are more likely to manifest tolerant attitudes toward IPV [15,17]. Attitudinal acceptance can further influence survivors’ help-seeking behavior; upon withstanding violence, IPV-tolerant survivors may be less likely to disclose or report abuse, as well as engage in help-seeking behaviors and escape violent relationships [18,19]. From a community perspective, attitudinal acceptance of IPV has been linked with a paucity of accessible victim support services as concomitant inequitable gender norms hinder the prioritization of GBV resources, programs, and health services [20]. Recent work by Garcia-Diaz et al. (2018) and Nagamatsu et al. (2016) provides valuable insight into these dynamics, suggesting that ascription to social norms advancing gender inequality and IPV tolerance undermines survivors’ ability to recognize GBV [21,22].

While the link between such experiences of IPV and adverse mental health outcomes has received extensive study, the relationship between attitudinal acceptance of IPV and mental health distress is less clear. In its 2017 white paper, the International Center for Research on Women (ICRW) reported the effect of unequal gender norms, and the role conflict they cause in adolescents, on augmenting suicidal behavior risks [23]. Furthermore, a wide body of literature evidences a negative correlation between IPV tolerance and help-seeking behavior among GBV survivors [24], commonly attributed to fear of stigma and reprisals from partners and family members due to disclosure [25,26,27]. Together, these trends support the possibility that attitudinal acceptance of IPV might modify the relationship between IPV victimization and mental health outcomes and suggest two hypotheses. In line with the Cry of Pain and SAMS models detailed by Panagioti et al. (2012) [12], IPV-tolerant GBV survivors may anticipate negative future outcomes due to feelings of defeat, leading to increased symptoms of mental distress and suicidal behavior. IPV-tolerant survivors may also be more likely to feel they deserved the violence and engage in self-blame, which has been associated with reduced self-esteem [28]. As such, interventions promoting IPV intolerance may foster hope and confidence, alleviating the sense of entrapment experienced by IPV survivors. Conversely, IPV-intolerant adolescent girls subjected to violence might experience a heightened sense of failure and entrapment, exhibiting a more profound deterioration in self-efficacy due to the incongruency between their predicament and values. This cognitive dissonance may elevate their risk of developing or aggravating mental health symptoms and suicidal behaviors. To add to the evidence on this topic, this multi-country study examines the relationship between attitudinal acceptance and adverse mental health outcomes in adolescent girls aged 13 to 24 exposed to IPV in sub-Saharan Africa.

## 2. Methods

### 2.1. Data

This analysis employed data from the Violence Against Children Surveys (VACS) conducted in Nigeria (2014), Malawi (2013), and Uganda (2015). The VACS datasets are implemented as part of a partnership between UNICEF, country governments, and the U.S. Centers for Disease Control and Prevention’s (CDC) Division of Violence Prevention, among other bilateral and multilateral organizations, and aim to estimate the scope, determinants and impacts of violence against children and adolescents. The VACS utilize a multi-stage sampling design and samples are selected to be representative of 13–24-year-old males and females in each country at the national level [29]. In order to reduce the possibility that both a survivor and perpetrator of the same incident of IPV are interviewed, the primary sampling units (PSUs) in the first stage are stratified by sex such that all PSUs are either all female or all male. In the second stage, approximately 20–25 respondents are randomly selected within each PSU. This analysis employs data from females only.

Gender-matched surveys—whereby male and female respondents were interviewed by male and female interviewers, respectively—for the three VACS included in this analysis were administered in-person. Survey questionnaires included questions on basic demographics; experiences of physical, sexual, and emotional violence; service access and utilization; and mental and sexual health, among other topics. Caregivers’ informed consent was obtained for participants under 18, and informed consent and assent were obtained directly from participants 18 and over and under 18, respectively. All interviews took place in a private space and did not involve the collection of any identifying information in order to protect respondents’ privacy and ensure confidentiality. All study protocols were approved by the CDC and in-country institutional review boards.

### 2.2. Measures of Interest

This analysis included two mental health outcomes of interest: suicide ideation and the Kessler 6 scale (K6). Suicide ideation was operationalized as a dichotomous outcome using the question, “Have you ever thought about killing yourself?” The K6 can be used to measure nonspecific distress and has been previously validated in adolescent populations [30,31]. The scale is derived from six questions asking about the frequency of symptoms over the past month, including nervousness, hopelessness, restlessness, feeling so sad that nothing can cheer you up, feeling that everything is an effort, and worthlessness. Respondents indicate whether they have experienced each symptom all the time, most of the time, some of the time, a little of the time, or none of the time in the past 30 days. The final K6 score assumes a value from 0 to 24, where lower scores signal greater mental distress. The Cronbach’s alpha for the K6 in this sample is 0.87.

The analysis also employs two predictors of interest. IPV victimization is operationalized in the binary and signals whether a respondent has ever experienced physical IPV by a current or former partner. A dichotomous measure of attitudinal acceptance of IPV is generated from five questions capturing whether a respondent believes it is acceptable for a husband to beat his wife in each of five scenarios: the wife goes out without telling her husband, does not take care of her children, argues with him, refuses to have sex with him, or burns the food. Respondents receive a “1” if they agree that IPV is an acceptable response in at least one scenario, and a “0” otherwise.

Control variables included in the regression analyses were selected according to the existing literature and the VACS questionnaires, and include age, a dichotomous variable signaling whether the respondent had ever attended school, a dichotomous variable signaling whether the respondent is married or living with someone as if married, and a dichotomous variable indicating whether the respondent has worked for pay in the past 12 months [32,33,34].

### 2.3. Analysis

All measures were first summarized to assess descriptive statistics within each country. Multivariate logistic and ordinary least square (OLS) regressions were used to estimate the relationships between the predictors of interest and suicide ideation and the K6 scale, respectively. In order to examine whether attitudinal acceptance of IPV modifies the relationship between violence victimization and mental health outcomes, an interaction term between attitudinal acceptance of IPV and violence exposure was included in the models. The regression analyses were stratified by country and all models controlled for age, ever school attendance, ever being married, and working for pay in the past 12 months. Observations were weighted to be representative of 13–24-year-old females in each of the three countries, and standard errors were adjusted to account for the multi-stage sampling design. All analyses were carried out using Stata14.

## 3. Results

Table 1 summarizes descriptive statistics. The average age of cohorts across all three countries was approximately 18 years. Ever school attendance was lowest in Nigeria, at 77.6%, compared to 95.1% in Malawi and 96.1% in Uganda. Labor force participation in the past year was also less common in Nigeria, where 51.9% of girls reported working in the past year, as compared to 79.7% in Malawi and 79.7% in Uganda. Violence exposure varied across contexts, with approximately 15% of girls in Uganda and Malawi reporting ever exposure to IPV, compared to 8.9% in Nigeria. Attitudinal acceptance of IPV was highest in Uganda, with 79.8% of girls expressing IPV-tolerant views, as compared to 59.5% in Nigeria and 66.9% in Malawi. Approximately 12.1% of girls in Uganda reported ever suicide ideation, compared to 6.3% and 5.7% in Nigeria and Malawi, respectively. On average, respondents exhibited comparable levels of mental health distress as determined by the K6 scale.

Table 2 presents findings from the regression analyses. Controlling for age, education, marital status, and work history, exposure to IPV was a significant predictor of adverse mental health outcomes as assessed by the Kessler 6 Scale in all three countries. Girls who reported being exposed to IPV in Nigeria and Uganda demonstrated 9.18 (CI 95% = [3.49, 24.14]; *p* = 0.001) and 4.36 (CI 95% = [2.09, 9.12]; *p* = 0.001) greater odds of suicide ideation, respectively.

Attitudinal acceptance of IPV was not found to be associated with either mental health outcome in any of the countries. However, results showed that among respondents who had experienced IPV, those who expressed attitudinal acceptance of IPV had significantly greater mental health in Nigeria (B = 2.86; CI 95% = [0.76, 4.96]; *p* = 0.01) and Malawi (B = 4.38; CI 95% = [ 0.95, 7.81]; *p* = 0.05) as measured by the Kessler 6 scale. IPV survivors who exhibited attitudinal acceptance of IPV also had significantly lower odds (aOR = 0.24; CI 95% = [0.07, 0.80]; *p* = 0.05) of reporting suicide ideation compared to their peers who did not accept IPV.

## 4. Discussion

Although existing literature finds a positive correlation between gender inequitable or IPV-accepting attitudes and suicidal behaviors, few studies have explored whether these relationships differ for women and girls who have and have not experienced IPV [23]. Previous evidence suggests that IPV survivors who believe IPV is acceptable may exhibit differential behaviors than their IPV-intolerant counterparts—such as one study which finds IPV survivors who endorse IPV are less likely to seek GBV services [35,36]. Yet, more evidence is needed on how these interactions influence outcomes of well-being. This study explores the impact of attitudinal acceptance of IPV on adverse mental health outcomes in IPV-exposed adolescent girls aged 13 to 24 in sub-Saharan Africa. Our analysis reveals that, among adolescent girls exposed to IPV in Nigeria and Malawi, those who expressed attitudinal acceptance of IPV exhibited decreased symptoms of mental distress as defined by the Kessler 6 Scale, compared to survivors who did not exhibit attitudinal acceptance of IPV.

One potential hypothesis for these findings may be related to the fact that survivors of IPV who do not consider IPV acceptable will consequently experience a discrepancy between their beliefs and reality [37]. This incongruency may breed cognitive dissonance, which has been shown to contribute to adverse mental health outcomes [38]. In one study on generational dissonance, Pottie et al. [39] show an association between cognitive discordance and depressive symptoms in adolescent immigrants [39]. Likewise, in their analysis, Dengah et al. (2019) attribute mental health symptomology, including depression, anxiety, and suicidality, in Mormon undergraduates to psychological stress caused by the dissonance between held gender roles and religious beliefs [40]. Alleviation of such psychological strain may be achieved by an attitudinal shift to elicit congruency in cognition. Thus, previous studies posit that IPV survivors’ denial of abuse or rationalization or minimization of violence serve as coping mechanisms to reduce stress triggered by attitudinal non-acceptance of IPV clashing with IPV-tolerant behaviors and staying decisions in abusive relationships [1,41].

On the other hand, as evidenced by Mitchell et al. [36], increased depressive symptoms have been observed in IPV survivors who accept responsibility for their abusive relationships. Such dynamics of self-blame may be more present among IPV intolerant survivors unable to rationalize their staying behaviors—providing another understanding for our findings of reduced mental health symptoms and suicide ideation in survivors who exhibit attitudinal acceptance of IPV. However, while attitudinal acceptance of IPV for survivors is correlated with reduced mental health symptomology, its use as an effective coping mechanism is precluded by its association with reduced help-seeking behavior.

While practitioners should certainly not consider promoting IPV tolerance as a coping technique for survivors in IPV interventions, they should account for the nuances of survivors’ coping strategies and ensuing ramifications of context-blind programming design. Namely, IPV interventions devoted to altering stakeholders’ and communities’ normative acceptance of IPV should simultaneously employ mental health and psychosocial support services (MHPSS) to evaluate and address the consequences of the cognitive dissonance that may result for women and girls who continue to experience IPV. Programs can work with survivors to identify emotion-focused coping behaviors, involving minimization, rationalization, and denial, and discuss how such strategies shape the cognitive appraisal frameworks prompting IPV tolerance in their relationships [36]. Subsequent interventions teaching problem-focused, active coping mechanisms, while targeting communities’ normative acceptance of violence and GBV resources, may reduce the mental distress experienced by survivors. “Problem-focused coping”, defined as active behaviors used to change stress-causing circumstances, including information gathering, help-seeking, and problem-solving, is associated with decreased depressive and PTSD symptoms and increased self-esteem and self-efficacy, especially in the setting of increased social support and congruent behavioral advice from their communities [41,42]. 

It is of note that significant improvement in mental health status and reduction in odds of suicide ideation for IPV-tolerant IPV survivors was only observed in the Nigerian and Malawian cohort. The lack of similar findings in Uganda suggests a need to further explore this dynamic in other sub-Saharan African countries and LMICs internationally. Replication of our findings in other cohorts, accompanied by improved characterization of context-specific sociocultural factors and respondent demographics, may provide insight for future identification of settings requiring mental health-sensitive modifications to GBV interventional approaches. Such examination could also aid in elucidating context-specific elements mitigating the role of IPV tolerance and cognitive dissonance in survivors’ mental health presentation in Uganda, providing novel targets for GBV program design.

Consideration of findings from this analysis should take into account the following limitations. First, the VACS are self-reported questionnaires, and thus measures of sensitive items such as suicide ideation and violence exposure may be susceptible to disclosure bias [25,26,27]. Second, our measure of mental well-being was constructed using only six survey questions and may not serve as a robust measure of mental health. Future research might consider replicating these findings with other clinical measures of mental health outcomes.

## 5. Conclusions

Attitudinal acceptance of IPV is associated with worsened depressive symptoms amongst IPV survivors, yet may provide an emotion-focused coping mechanism to assuage the cognitive dissonance experienced by survivors. Assessing respondents’ coping strategies, and risks of mental health distress posed by interventions, is critical for mitigating potential unintended consequences of IPV interventions and ensuring the long-term success of programs. Our research demonstrates a context-specific, mediating role for IPV tolerance in promoting improved mental health status and reduced suicide ideation in IPV-exposed adolescent girls. The analysis thus highlights the importance of a nuanced approach to IPV interventions to reduce symptoms of mental health distress in survivors through the targeting of context-specific sociocultural factors and norms.

## Figures and Tables

**Table 1 ijerph-18-05099-t001:** Descriptive statistics of adolescent girls aged 13–24 years.

	Nigeria	Malawi	Uganda
Age, Mean (SD)	18.43 [3.60]	18.27 [3.55]	18.31 [3.40]
Ever Attended School	77.6	95.1	96.1
Worked in Past Year	51.9	79.7	79.7
Married	30.3	39.6	36.5
Experienced IPV	8.9	14.6	15.6
IPV-Tolerant	59.5	66.9	79.8
Kessler 6 scale	20.37 [4.78]	20.06 [4.47]	19.06 [4.95]
Suicide Ideation Sample Size (n)	6.3 1766	5.7 1029	12.1 3159

Note: Data are represented as percentages or, in the case of age and the K6 scale, as means [SD]. Respondents were considered married if they were in a formal marriage or living with a partner as if married. All observations were weighted to ensure representativeness of the population of 13–24-year-old females in each country.

**Table 2 ijerph-18-05099-t002:** Examining the effect of attitudinal acceptance of IPV on mental health symptoms, beta coefficient (B), and adjusted odds ratios (aOR).

	Kessler Scale B [CI 95%]	Suicide Ideation aOR [CI 95%]
**Nigeria**		
Ever exposure	−3.91 *** [−5.82, −2.01]	9.18 *** [3.49, 24.14]
IPV-tolerant	−0.49 [−1.11, 0.14]	1.77 [0.93, 3.36]
Ever exposure *IPV-tolerant	2.86 ** [0.76, 4.96]	0.24 * [0.07, 0.80]
R-squared	0.042	0.073
**Malawi**		
Ever exposure	−5.46 *** [−7.67, −3.24]	1.40 [0.27, 7.25]
IPV-tolerant	−0.05 [−0.66, 0.56]	0.69 [0.38, 1.27]
Ever exposure *IPV-tolerant	4.38 * [0.95, 7.81]	1.18 [0.19, 7.43]
R-squared	0.102	0.083
**Uganda**		
Ever exposure	−3.61 ** [−5.96, −1.26]	4.36 *** [2.09, 9.12]
IPV-tolerant	−0.38 [−1.38, 0.62]	1.18 [0.50, 2.75]
Ever exposure *IPV-tolerant	1.80 [−0.74, 4.34]	0.64 [0.27, 1.54]
R-squared	0.085	0.080

Note: ordinary least squares regressions are used to estimate the Kessler 6 scale, and logistic regressions are used to estimate the incidence of suicide ideation. Confidence intervals (CIs) are expressed at 95%. Beta coefficients are presented for Kessler Scale estimates, and adjusted odds ratios are displayed for suicide ideation estimates. Data are weighted to reflect the 13–24-year-old female population in each country, and standard errors are adjusted for complex sampling design. All models control for age, education, marital status, and past year’s work history. aOR = adjusted odds ratio. B = beta coefficient. Lower scores on the Kessler 6 scale signal greater mental distress. Odds ratios and coefficients are statistically significant at * *p* < 0.05, ** *p* < 0.01, and *** *p* < 0.001.

## Data Availability

Data are available upon request at: https://www.togetherforgirls.org/request-access-vacs/ (accessed on 1 July 2018).

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
