# Peer review of "Attitudinal Acceptance of Intimate Partner Violence and Mental Health Outcomes for Female Survivors in Sub-Saharan Africa"

_ijerph, 2021, doi:10.3390/ijerph18105099_

Round 1

Reviewer 1 Report

Overall, this is a well-written paper that explores an interesting research question and utilizes existing evidence and theoretical literature well. My main critique is the use and operationalization of the sexual violence measure – see Methods below.

A small point regarding language: throughout the terminology on mental health is unusual and could be phrased better – for example the phrase “mental health disturbance” is not utilized widely in the literature. Please consider replacing with either “poor or adverse mental health outcomes” or “increased symptoms of mental distress.” Other phrases used to indicate higher levels of mental distress are also inaccurate or confusing – mental health exacerbation, greater mental health. This often leads to the results being unclear – i.e. reduced mental health exacerbation. Please review the manuscript for these terms and replace for clarity and consistency.

Introduction:

  1. Please update the WHO citation/ data in the first paragraph to the 2021 estimates to ensure up to date data;

  1. Paragraph on IPV tolerance: this paragraph could use a brief definition of IPV tolerance, and a few sentences more generally on levels of IPV acceptance globally/ in Sub-Saharan Africa/ in these countries – before going into the associations with IPV victimization and perpetration

Methods:

  1. How and why did you select these control variables in the model? Are there others that were considered and not included?

  1. The sexual violence measure is not a measure of IPV but of any kind of sexual violence, which may include child sexual abuse and/ or IPV. The conceptual model that is proposed and the literature that is drawn on is all focus on IPV, but the literature on child sexual abuse and mental health is quite different, and the associations with IPV acceptance may be different or less relevant for a survivor of child sexual abuse. I suggest ideally focusing only on sexual IPV (if the data allows analysis by perpetrator). If this is not possible from the data, then I would recommend dropping the sexual violence analysis or adding significant literature and recognition of the differences between physical IPV and sexual violence (IPV and other).

Discussion:

  • Can you explore if or how the cross-sectional nature of the data impacts your findings and interpretation? Could IPV attitudes pre-date IPV exposure, and if so, how is this relevant to your interpretation?

Conclusion:

  • The first sentence indicates finding on help-seeking behaviours but these data do not appear to be shown in the paper

Reviewer 2 Report

Comments:

1.- The references section and citations in the manuscript should be adapted to IJERPH standards.

2.- IPV variable is defined as physical violence by an intimate partner (current or past). However, it is not clear whether the sexual violence is carried out by the partner or not. I consider it relevant to point out this feature.

3.- The variable "attitudinal acceptance of IPV" refers only to physical violence by an intimate partner. Therefore, it is not clear its relevance as a predictor of sexual violence, especially if it is not carried out by the intimate partner (see comment 2). I believe that greater theoretical justification of why the acceptance of IPV may affect the consequences of sexual violence would be necessary.

4.- It would be advisable to include, in Table 2, some measure of goodness of fit of the models; without this information, although some coefficients are statistically significant, it is difficult to assess the impact of the predictor variables on the endogenous ones.

5.- Since, Table 2 shows that tolerance IPV was not statistically significant as a predictor of sexual violence in any of the three cohorts; the main objective of the manuscript is to analyze their influence on the mental consequences of violence; and previous comments in paragraphs 2 and 3; I think it would be advisable further justification for the inclusion of sexual violence models or simply disposal of the manuscript, focusing only in the case of IPV violence.

6.- In Table 2, for “Ever exposure * IPV tolerant” from the Malawi cohort, the reported aOR is 1.18 with [0.19, 7.43] as 95% CI, but it is not marked as statistically significant. I think this could be a simple asterisk omission, so I would recommend reviewing it.
